# Effect of Dynamic High-Pressure Microfluidizer on Physicochemical and Microstructural Properties of Whole-Grain Oat Pulp

**DOI:** 10.3390/foods12142747

**Published:** 2023-07-19

**Authors:** Ping Jiang, Ziyue Kang, Su Zhao, Ning Meng, Ming Liu, Bin Tan

**Affiliations:** Institute of Cereal and Oil Science and Technology, Academy of National Food and Strategic Reserves Administration, Beijing 100037, China; jp@ags.ac.cn (P.J.); kang_ziyue@163.com (Z.K.); zhaosuzs@163.com (S.Z.); lm@ags.ac.cn (M.L.)

**Keywords:** whole-grain oat pulp, dynamic high-pressure microfluidizer, stability, nutritional value, microstructure

## Abstract

By avoiding the filtration step and utilizing the whole components of oats, the highest utilization rate of raw materials, improving the nutritional value of products and reducing environmental pollution, can be achieved in the production of whole-grain oat drinks. This study innovatively introduced a dynamic high-pressure microfluidizer (DHPM) into the processing of whole-grain oat pulp, which aimed to achieve the efficient crushing, homogenizing and emulsification of starch, dietary fiber and other substances. Due to DHPM processing, the instability index and slope value were reduced, whereas the β-glucan content, soluble protein content and soluble dietary fiber content were increased. In the samples treated with a pressure of 120 MPa and 150 MPa, 59% and 67% more β-glucan content was released, respectively. The soluble dietary fiber content in the samples treated with a pressure of 120 MPa and 150 MPa was increased by 44.8% and 43.2%, respectively, compared with the sample treated with a pressure of 0 MPa. From the perspective of the relative stability of the sample and nutrient enhancement, the processing pressure of 120 MPa was a good choice. In addition, DHPM processing effectively reduced the average particle size and the relaxation time of the water molecules of whole-grain oat pulp, whereas it increased the apparent viscosity of whole-grain oat pulp; all of the above changes alleviated the gravitational subsidence of particles to a certain extent, and thus the overall stability of the system was improved. Furthermore, CLSM and AFM showed that the samples OM-120 and OM-150 had a more uniform and stable structural system as a whole. This study could provide theoretical guidance for the development of a whole-grain oat drink with improved quality and consistency.

## 1. Introduction

Oats (*Avena sativa* L.) are annual herbs belonging to the genus Avena of the grass family and are a crop cultivated worldwide. Their main producing areas are temperate regions of the Northern Hemisphere, such as Russia, Canada, the United States, Australia, Germany, Finland and China, which rank as the top seven in the world in terms of grain output. Oats are one of the eight major human food crops [1,2]. China’s oat planting area and yield are relatively stable. In 2022, China’s oat planting area was 530,000 hectares, with a yield of 640,000 tons. Compared with other cereals, oatmeal has outstanding nutritional and health functional benefits, and its main nutritional components are more comprehensive [3]. In addition to rich proteins, polysaccharides and lipids, it is also rich in lysine, β-glucan, unsaturated fatty acids, phenols, etc., playing an important role in lowering cholesterol, regulating intestinal flora diversity, promoting fat metabolism and so on [4,5,6]. It is known as the “king of the nine kinds of grains” and is a high-quality grain that can be used as a “dietary therapy”. It ranks third out of “China’s Ten Best Whole Grains”.

Cereal beverages represent a new direction of cereal processing. This is one of the current hot spots of oatmeal product development, where the traditional edible form has been changed into a liquid state. Compared with milk, oat drinks are environmentally friendly, economize resources, and conform to the concepts of green, low-carbon, environmental and sustainable development [7,8]. Many countries have developed pure cereal drinks using oats as raw materials, such as European countries, the United States and Japan [9]. With the continuous enhancement of health awareness and environmental awareness in Chinese consumers, the development of oat drink processing technology and product research in China has been very rapid in the past decade, and new oat drinks keep emerging. However, the traditional preparation process of oat milk mainly includes enzymatic hydrolysis, colloid grinding, filtration, homogenization and filling, where about 1 kg of rolled oat produces 2.85 kg of oat milk [10,11]. In the filtration step, about 20% of the filter residue is generated, which is usually discarded, resulting in environmental pollution and a large amount of nutrient waste. Therefore, how to effectively utilize this oat filter residue is a difficult problem in the industrial production of oat milk. Using the entire oat may be an alternative method. Whole-grain oat drinks avoid the filtration step and use all of the oat components, achieving the highest utilization rate of raw materials and reducing environmental pollution. However, there is little research about whole-grain oat pulp at present, and only Li et al. [12] produced a spontaneously stable whole soybean milk using an industry-scale micofluidizer system without soaking soybeans and filtering residues. The reason for this is that the existence of starch and dietary fiber in whole-grain oat pulp causes a rough taste and poor stability. Therefore, efficient crushing and homogenizing emulsification technology is the key to developing whole-grain oat pulp.

A dynamic high-pressure microfluidizer (DHPM) is a machine that subjects multiphase flow materials to the triple action of a high-speed shear effect, high-pressure-jet to offset the impact of energy, and a hole effect generated by the instantaneous flow channel pressure difference [13]. High-efficiency ultrafine grinding (particle size from tens of microns to several microns), homogenization, emulsification (emulsion particle size up to the nanometer-level narrow peak distribution) and sterilization can lead to the degradation of macromolecular polymers; a change in the gel hardness, elasticity, cohesiveness, adhesiveness and chewability of starch; and the degradation of non-starch polysaccharides [14,15,16]. These processes can also change the apparent viscosity, gel properties, water absorption, water retention, fluidity, particle size and other properties.

Therefore, this study innovatively introduced dynamic high-pressure microfluidizer technology for the processing of whole-grain oats to prepare a kind of whole-grain oat pulp that can be self-stabilized without adding any stabilizer or emulsifier. The stability mechanism is preliminarily discussed in order to solve the problems of the low utilization rate of whole components, utilizing clean and slag-free processing technology in cereal drinks, which has important implications for promoting the consumption of whole-grain oats in practice. The results of this study can provide a processing technology for the preparation of whole-grain oat pulp in the food industry using an industry-scale microfluidizer system.

## 2. Materials and Methods

### 2.1. Materials

Whole-grain oats were obtained from the Zhangjiakou Academy of Agricultural Sciences. Bovine serum albumin (BSA, >98% pure) was purchased from Sigma-Aldrich Chemical Ltd. (Saint Louis, MO, USA). Rhodamine B, Calcofluor White, fluorescein 5-isothiocyanate (FITC) and Nile Red were purchased from Sigma-Aldrich Chemical Ltd. (Shanghai, China). The β-glucan assay kit, damaged starch assay kit and dietary fiber assay kit were purchased from Megazyme Co. (Bray, Wicklow, Ireland). 

### 2.2. Preparation of Whole-Grain Oat Pulp

The whole-grain oats were treated with distilled water (100 °C) at a total ratio of 1:10 (oat: water, *w*/*w*) for 3 min. After the above treatment, the oats were taken out and cooled to room temperature (25 °C), mixed with distilled water at a ratio of 1:10 (*w*/*w*), and precrushed in the wet grinder (XCFG-2018, Beijing Collaborative Innovation Food Technology Co., Ltd., Beijing, China) to prepare the whole-grain oat pulp. The frequencies of the primary and secondary crushing mills of the pre-pulverizer were both 40 Hz, and 12.5 and 37.5 Hz for the water pump and screw feeder, respectively. Then, the whole-grain oat pulp was transferred into the dynamic high-pressure microfluidizer (LM10, Microfluidics International Corporation, Newton, MA, USA) and the treatment pressures were set as 60, 90, 120 and 150 MPa. The samples were labeled as OM-60, OM-90, OM-120 and OM-150, whereas the sample that only underwent wet grinding was labeled as OM-0.

### 2.3. Dispersion Characteristics of a Stable System

The measurements were performed using a concentration system dispersion stability analyzer (LUMiSizer-611, L.U.M. GmbH, Berlin, Germany), according to Tai et al. [17]. The samples were scanned every 10 s for 1 h at 25 °C at a speed of 4000 rpm. The instrument automatically recorded the scanning results and calculated the instability index and slope value of the samples.

### 2.4. Particle Characteristics

The particle sizes and distributions of the samples were determined using a laser diffraction particle size analyzer (Mastersizer 2000, Malvern Instruments Co., Ltd., Worcestershire, UK), using the method of Lu et al. [18] with little modification. The specific parameters were set as follows: particle refraction index of 1.59, particle absorption rate of 0.001 and deionized water as the dispersant (refractive index of 1.330). ‘Span’ was used to determine the distribution width of droplet sizes through the following formula:Span = [D(0.9) − D(0.1)]/D(0.5)

In this formula, D(0.9), D(0.1) and D(0.5) represent diameters at a 10%, 50% and 90% cumulative volume, respectively.

### 2.5. Typical Nutrient Content of Whole-Grain Oat Pulp

#### 2.5.1. Determination of β-Glucan Content

The β-glucan content was determined using the AACC (Method 32-23.01), using a β-glucan assay kit (Megazyme, Wicklow, Ireland).

#### 2.5.2. Determination of Soluble Protein Content

The soluble protein content of the samples was determined via the method of Galdeano et al. [19], with slight modifications. In total, 10 mL of each sample was centrifuged at 8000× *g* rpm for 20 min, and 0.5 mL of the supernatant was diluted 10 times and dyed with Coomassie blue G250 solution. The absorbance was 595 nm. Bovine serum albumin (BSA, >98% pure, Sigma, Ronkonkoma, NY, USA) was used as a standard protein.

#### 2.5.3. Determination of Insoluble and Soluble Dietary Fiber Content

The dietary fiber content was determined using a dietary fiber assay kit (Megazyme, Wicklow, Ireland).

### 2.6. Rheological Properties

The rheological properties were determined according to the method described by Sharafbafi et al. [20] with slight modifications, which included using a rheometer (AR2000, TA Instruments, Newcastle, UK) with a 40 mm diameter cone plate, and the rheological properties were measured with a shear rate that ranged from 0.1 to 100 s^−1^ at 25 °C. The apparent viscosity profiles were detected.

### 2.7. Water Relaxation Properties

The water relaxation properties of the whole-grain oat pulp treated with different pressures were determined via low-field nuclear magnetic resonance (LF-NMR) (NMI20-040 V-I, Shanghai Niumay Electronic Technology Co., Ltd., Shanghai, China), using the method of Patra et al. [21] with a few modifications. In total, 5 mL of the sample was poured into an LF-NMR detection tube and detected in a resonance detector at 20 °C. The transverse relaxation time constant (T_2_) was measured using a CPMG pulse sequence with a 4 s relaxation delay, 12,000 echoes, 8 scans, and 5 in gain, where τ was set to 400 μs.

### 2.8. Confocal Laser Scanning Microscope Analysis

The microstructure of the sample was analyzed via confocal laser scanning microscopy (CLSM, LSM710 NLO, Zeiss, Gottingen, Germany), using the method of Huang et al. [7] with a few modifications. All of the samples were stained with 0.025% *w*/*v* Rhodamine B, 0.01% *v*/*v* Calcofluor White, 0.25% *w*/*v* fluorescein 5-isothiocyanate (FITC) and 0.1% *w*/*v* Nile Red. The final concentration of each dye in the sample was around 0.01 μg dye/mL. Then, 50 μL of the dyeing sample was placed in a slide for observation. Images were taken using a CLSM multiphoton system with a 40 × objective lens. The excitation–emission wavelengths were at 637 nm for Rhodamine B, 521 nm for FITC, 445 nm for Calcofluor White and 568 nm for Nile Red.

### 2.9. Atomic Force Microscope (AFM) Analysis

AFM imaging was performed in tapping mode using a silicon probe (NT-MDT Prima; Bruker Dimension Edge, OLS 5000, Olympus, Japan) with a nominal spring constant k = 42 N∙m^−1^ and a nominal tip radius R = 8 nm, using the method of Verran et al. [22] with a few modifications. 

### 2.10. Statistical Analysis

All experiments were performed in triplicate. Significant differences were calculated using SPSS Statistics 20.0 (SPSS, Chicago, IL, USA), and data were analyzed using one-way ANOVA and a post hoc Duncan’s test for multiple comparisons. All data are presented as means ± SEM. The figures were prepared using GraphPad Prism (GraphPad Software, San Diego, CA, USA).

## 3. Results and Discussion

### 3.1. Stable System Dispersion Characteristics of Whole-Grain Oat Pulp

The stability of the samples was compared using the instability index and slope value, wherein the instability index refers to the change in the light transmittance of each part of the sample over time during centrifugation, and the slope value (%/s) refers to the slope of the light transmittance integral curve over time. The smaller the instability index is, the less stratified the sample is and the better the stability is [23]. The larger the slope value is, the faster the transmittance of the sample changes within a certain period of time, which means that the faster the moving delamination velocity of the sample changes, the more unstable the sample is. 

It has been reported that high pressure microfluidization can enhance the physical stability of ketchup-type products [24] and improve the quality of Ottoman Strawberry (*F. Ananassa*) juice [25]. As is shown in Figure 1, with the increase in the treatment pressure, the instability index and slope value of the whole-grain oat pulp became smaller, which indicates that the stability of the samples increased. The variation range in the instability index and the slope value of the samples with a treatment pressure greater than 120 MPa became inconspicuous.

### 3.2. Typical Nutrient Content of Whole-Grain Oat Pulp

The effects of different pressures on β-glucan content and soluble protein content in whole-grain oat pulp are shown in Figure 2a. The β-glucan content of different oat varieties varied from 3.14% to 7.43%, and the maximum difference was 4.29% [26]. It can be seen that the β-glucan content of the whole-grain oat pulp gradually increased with the increase in the treatment pressure, but there was no significant difference in the β-glucan content when the treatment pressure was higher than 120 MPa. Compared with the sample labeled OM-0, the samples treated with 120 MPa and 150 MPa released 59% and 67% more β-glucan content, respectively. Kivelae et al. [27] reported that high-pressure homogenization will promote the breakage of intermolecular valence bonds, molecular cracking and molecular polarity change, resulting in more clustering and an increase in the gelation temperature, solubility, expansion, apparent viscosity and consistency coefficient of β-glucan in the product. As the treatment pressure increased, the increased breakage of the cell and cell wall caused more β-glucan to dissolve and more β-glucan and other substances to cluster [28]. Some studies have pointed out that β-glucan can improve the stability of cereal beverages [7,29]; so, it can be inferred that a high-pressure microfluidizer can effectively improve the stability of a whole-grain oat pulp system.

With the increase in treatment pressure, the soluble protein content in the whole-grain oat pulp showed a trend of firstly increasing to reach the maximum value when the pressure was 120 MPa, and then decreasing. There was no significant difference when the treatment pressure was higher than 90 MPa, as shown in Figure 2a. This may be the reason as to why the increase in pressure made the shear force stronger and why the degree of sample fragmentation was greater, and thus why more soluble protein was dissolved. Hu, Zhao, Sun, Zhao and Ren [30] indicated that the microfluidization at different pressure levels improved the solubility, emulsifying properties and surface hydrophobicity of peanut protein isolate. Microfluidization could also increase the foaming and emulsifying properties of ovalbumin by enhancing its flexibility and surface activities [31]. However, when the treatment pressure was too high, it meant that the shear force, compressive collision and rapid shock were more intense, which may cause protein oxidation, denaturation or disintegration because of the high-temperature effect and the destroyed structure [32]; thus, the change in the soluble protein content showed a downward trend in this context.

As shown in Figure 2b, the soluble dietary fiber content in the whole-grain oat pulp increased with the increasing pressure, and the total dietary fiber content experienced no significant change. Compared with the samples treated with 0 MPa, the soluble dietary fiber content in the samples treated with 60 MPa, 90 MPa, 120 MPa and 150 MPa increased by 14.7%, 19.9%, 44.8% and 43.2%, respectively. This may be due to the effects of the DHPM, such as high-speed impact, high-speed shear, the hole effect and high-speed oscillation, which broke and loosened the dense outer layer of the fiber and loosened the tissue. The force between the molecular chains of the fiber polymer polysaccharides weakened or disappeared and the degree of polymerization decreased, resulting in an increase in SDF content [33]. Tu et al. [34] also reported that microfluidization treatment could lead to the redistribution of dietary fiber from the insoluble fraction to the soluble one. To summarize, the DHPM is an effective technology which can transform insoluble substances into soluble substances when making whole-grain oat pulp, improving the stability of the system and improving its nutritional value.

### 3.3. Particle Characteristics

Particle characteristics are the main factors that influence the stability of liquid materials like emulsions and beverages [35,36]. It can be seen that the DHPM can effectively reduce the average particle size of the whole-grain oat pulp, as shown in Table 1. With the increase in treatment pressure, all particle size indices of the samples showed a decreasing trend. The D [4,3] value of the whole-grain oat pulp without high-pressure treatment was 72.597 μm, and it reduced to 28.176 μm, 22.776 μm, 19.542 μm and 18.608 μm after the 60 MPa, 90 MPa, 120 MPa and 150 MPa treatments, respectively. The whole-grain oat pulp particles were gradually broken under the action of highly dense energy generated by cavitation and collision [37]. The better the crushing effect and the smaller the particle size, the greater the pressure [38]. The values of D [3,2] and D [4,3] were not very close, indicating that the shape of the sample particles was not very regular and the particle size distribution was also not very concentrated, which was consistent with the changing trend of the Span value.

Figure 3 shows the effect of different high pressures on the particle size distribution of whole-grain oat pulp. There were two main peaks in the particle size distribution diagram. The left particle size peak ranged from 1 to 50 µm and represented the oil body–protein aggregates or protein [39,40]. The right particle size peak ranged from 50 to 300 µm, and mainly consisted of dietary fibers and starch or particle aggregates [41,42,43,44]. The main particles of the sample OM-0 were concentrated in sizes of about 126 µm and 28 µm (intermediate particle size). The particle size distribution gradually deviated to the left along with the increase in treatment pressure; the main particles of the samples were concentrated in about 60 µm and 9 µm values. This indicates that the DHPM has a good crushing effect on both large particles and small particles such as protein, dietary fiber and starch [15,30,45]. When the treatment pressure reached 120 MPa and 150 MPa, the boundary between the two peaks in the particle size distribution became blurred, which may be due to the partial repolymerization of small particles under a higher pressure.

### 3.4. Rheological Properties

Viscosity is the physical barrier for phase separation in whole-grain beverages. Rheological characteristics are essential for developing products with good taste and stability [46]. As shown in Figure 4, the apparent viscosities of the whole-grain oat pulp under different high pressures all showed an obvious shear-thinning phenomenon with the increase in shear rate, which indicated that the whole-grain oat pulp is a typical non-Newtonian pseudoplastic fluid [47]. In addition, the strong shear force, high-speed impact, high-frequency vibration and thermal effect generated during DHPM processing led to the apparent viscosity of whole-grain oat pulp gradually increasing with the increase in the treatment pressure, which was caused by the conformational change in proteins, the remodeling of the intermolecular force of fiber fragments, and the increase in the contents of soluble polysaccharides and soluble dietary fiber in the whole-grain oat pulp [48,49]. Thus, the increase in and maintenance of the viscosity of the system can alleviate the gravitational subsidence of particles to a certain extent, so that the overall stability of the whole-grain oat pulp system can be improved.

### 3.5. Water Relaxation Properties

The relaxation time reflects the molecular motility, molecular binding state and pore-size distribution of the samples. Therefore, this property can be used to analyze the stability of whole-grain beverages through nuclear magnetic relaxation spectra and relaxation time [21]. As can be seen from Figure 5 and Figure 6, with the increase in treatment pressure, the relaxation time of water molecules in the whole-grain oat pulp showed a gradually decreasing trend. Meanwhile, there was no significant difference in the relaxation time and the change rate of relaxation time in the whole-grain oat pulp when the treatment pressure was greater than 120 MPa. This may be due to the fact that an increase in treatment pressure leads to an decrease in the particle size in the system, and an increase in the contents of soluble polysaccharides, soluble dietary fiber and soluble protein, which generates more hydrophilic groups in the system [41]. Therefore, the binding force of water molecules in the system became stronger with the increased pressure, and thus the relaxation time of water molecules decreased and the change rate decreased as well.

### 3.6. Confocal Laser Scanning Microscope (CLSM) Analysis

The microstructures of the whole-grain oat pulp samples under different high-pressure treatments were observed using a typical CLSM analysis. The dark-green and bright-green zones stained with FITC represent starch and protein, respectively. Blue zones stained with fluorescent white represent the cell wall and β-glucan. Red zones stained with Nile Red represent oil. Purple zones stained with Rhodamine B represent the protein. The four single-colored images were overlapped to obtain the multicolor images.

As shown in Figure 7, the starch particles became smaller with the increase in treatment pressure (A1–E1) and the regional concentration of starch changed when the pressure reached 120 MPa. The low concentration is represented as a darker background region, like a black hole, while the high concentration is represented as a more fuzzy background region, like a cloud. The proteins in the whole-grain oat pulp sample treated with a pressure of 0 MPa were clustered together (A2). Along with the increase in the treatment pressure, free proteins gradually infiltrated into the whole-grain oat pulp system because the original intact protein particles cracked. The spherical proteins became soluble protein monomers or oligomers with a small particle size, and the soluble protein content in the system increased, showing flocculation or fog (A2–E2) [50]. 

The third row in Figure 7 shows the effect of DHPM processing on the structure and distribution of β-glucan in the whole-grain oat pulp system. It can be seen that a high pressure promoted the release of β-glucan in the system, and the system formed a network-like structure, which further explains why the system became more stable with the increase in treatment pressure. The oil body content in the whole-grain oat pulp was very low and the change in oil body particles was similar to that of protein. With the increase in treatment pressure, the size of the oil body decreased, and the binding effect between the oil body and protein body became stronger and stronger. Finally, oil body particles gradually disappeared and were evenly dispersed in the system, thus enhancing the stability of the system [12].

According to the multicolor image superimposed in Figure 7(A5–E5), it can be seen that the boundaries for the particles are less noticeable and the boundaries tend to connect. This result indicates that a gel-like structure was constructed, which is similar to the results from previous reports [51]. With the increase in processing pressure, the particle size of each substance in the whole-grain oat pulp system became smaller, and the interaction, mutual penetration and combination abilities between them became stronger; thus, the samples OM-120 and OM-150 showed a more uniform and stable structure as a whole.

### 3.7. Atomic Force Microscope (AFM) Analysis

Atomic force microscopy (AFM) imaging technology has been proven to be a versatile tool for analyzing multiphase materials, as it can study the surface morphology, nanostructure and chain conformation of samples to obtain information about material surface roughness and material surface defects, as well as track the surface structure morphology [52]. It can also produce a rich three-dimensional simulation display of the morphology of the samples, so that the image is more suitable for people’s intuitive vision. Thus, it is an efficient microscopic imaging technology for characterizing multiphase materials.

Figure 8 shows the analysis results of the 2D geometry and 3D height morphology of the whole-grain oat pulp samples prepared with different DHPM treatment pressures and using AFM imaging technology. It can be seen from the 2D main morphology diagram that the substances in the whole-grain oat pulp sample without DHPM treatment (OM-0) presented with a state of dispersed distribution. With the increase in treatment pressure, the intermolecular force between the substances in the system gradually increased, and the initially generated chain structure gradually formed an interwoven network structure when the treatment pressure reached 120 MPa. This result was also echoed by CLSM and further explained the reason why its stability increased with an increased treatment pressure at the microscale. As can be seen from the 3D main morphology diagram in Figure 8, the surface morphology and structure distribution of the whole-grain oat pulp samples became more uniform, and the surface roughness of the samples also became lower with the increase in treatment pressure.

## 4. Conclusions

The effect of the dynamic high-pressure microfluidizer (DHPM) on the physicochemical and microstructural properties of whole-grain oat pulp was investigated. Due to DHPM processing, the instability index and slope value were reduced, whereas the β-glucan content, soluble protein content and soluble dietary fiber content increased. In addition, DHPM processing could effectively reduce the average particle size in whole-grain oat pulp, although the shape of the sample particles was irregular and the particle size distribution was unconcentrated. The apparent viscosity of the whole-grain oat pulp increased and the relaxation time of water molecules decreased due to DHPM processing; the increase in the viscosity and the stronger binding force of water molecules in the system could alleviate the gravitational subsidence of particles to a certain extent, and thus the overall stability of the whole-grain oat pulp system was improved. Furthermore, CLSM and AFM also showed that the samples OM-120 and OM-150 had a more uniform and stable structure as a whole. Based on the specific changes in all of the above indicators, it was determined that a processing pressure of 120 MPa should be selected as the best process parameter for the preparation of whole-grain oat pulp in further studies. This study provides theoretical guidance for the development of whole-grain oat drinks with improved quality and consistency.

## Figures and Tables

**Figure 1 foods-12-02747-f001:**
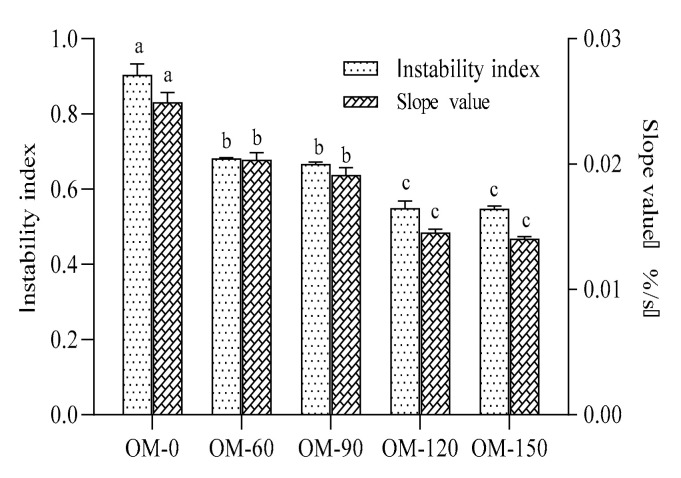
Effects of different DHPM treatment pressures on stability of whole-grain oat pulp. Different lowercase letters indicate significant differences among processing techniques (*p* < 0.05).

**Figure 2 foods-12-02747-f002:**
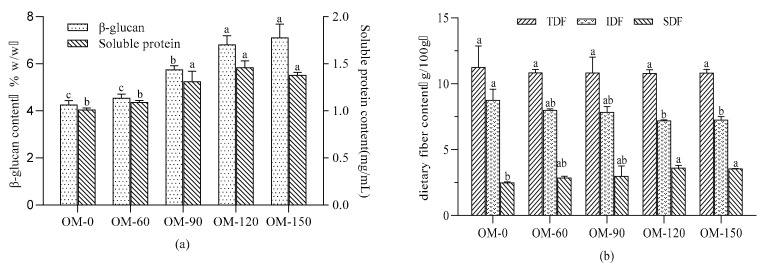
Effects of different DHPM treatment pressures on nutritional components of whole-grain oat pulp. (**a**) The content of β-glucan and soluble protein. (**b**) The content of total dietary fiber, insoluble dietary fiber and soluble dietary fiber. Different lowercase letters indicate significant differences among processing techniques (*p* < 0.05).

**Figure 3 foods-12-02747-f003:**
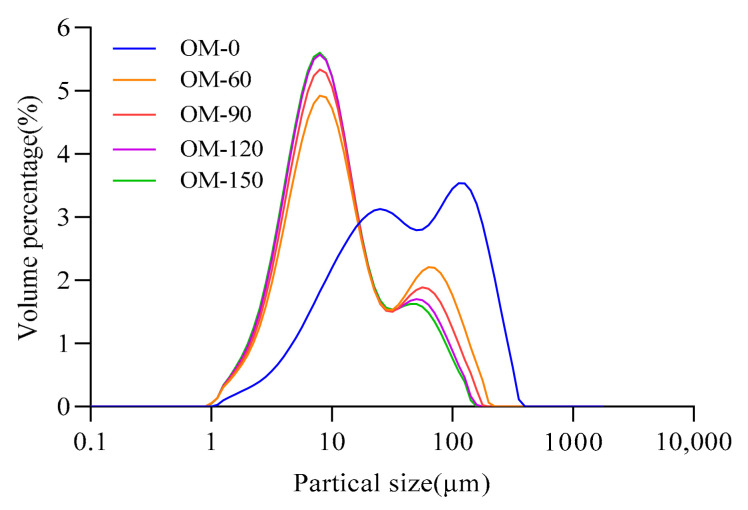
Effects of different DHPM treatment pressures on particle size distribution of whole-grain oat pulp.

**Figure 4 foods-12-02747-f004:**
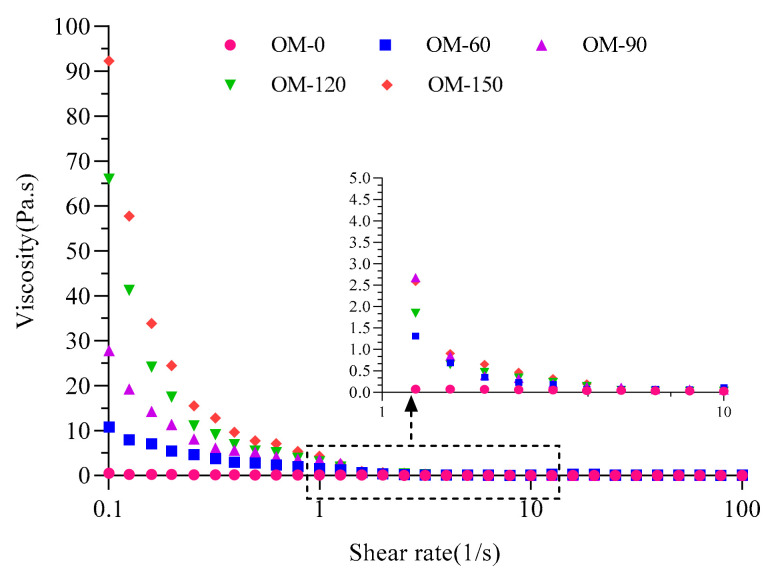
Effects of different DHPM treatment pressures on rheological properties of whole-grain oat pulp.

**Figure 5 foods-12-02747-f005:**
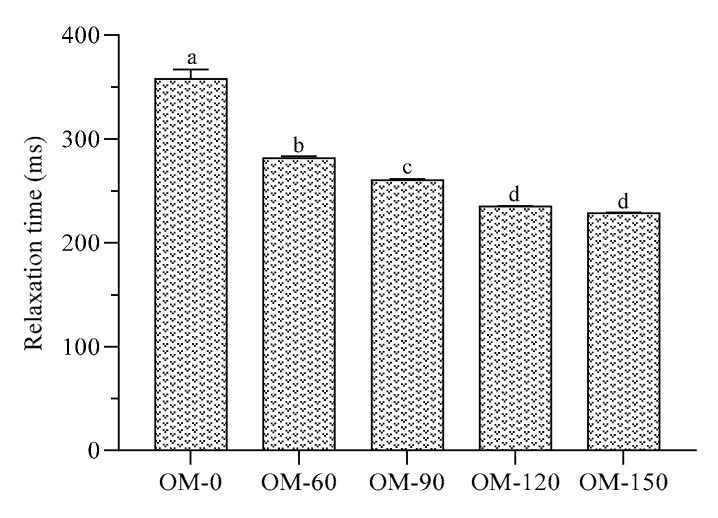
Relaxation time of whole-grain oat pulp under different DHPM treatment pressures. Different lowercase letters indicate significant differences among processing techniques (*p* < 0.05).

**Figure 6 foods-12-02747-f006:**
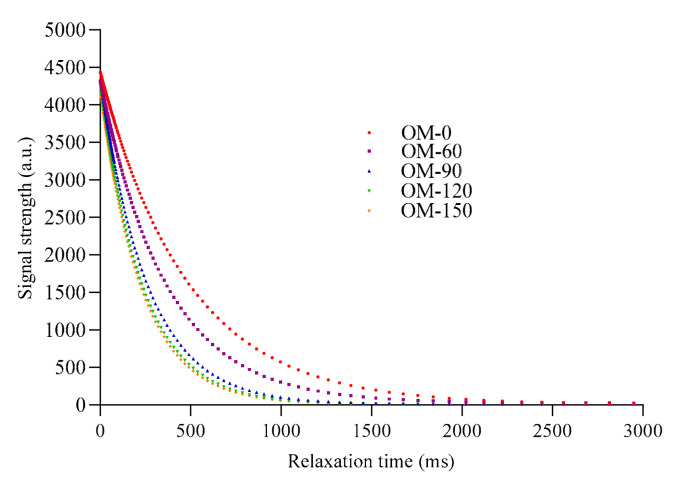
Relaxation time inversion diagram of whole-grain oat pulp under different DHPM treatment pressures.

**Figure 7 foods-12-02747-f007:**
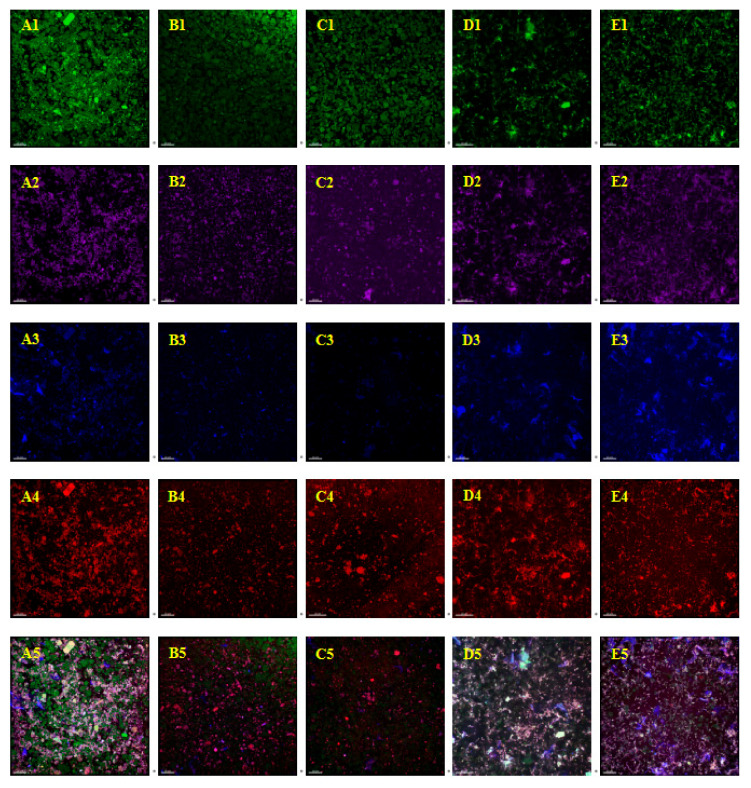
CLSM images of whole-grain oat pulp samples prepared at different pressures. (**A**): 0 MPa; (**B**): 60 MPa; (**C**): 90 MPa; (**D**): 120 MPa; (**E**): 150 MPa. Rows 1–4 represent the microscopic structure diagram stained with FITC, Rhodamine B, fluorescent whitening and Nile Red, respectively. The fifth row shows a polychromatic image of the previous four images in the row superimposed onto each other.

**Figure 8 foods-12-02747-f008:**
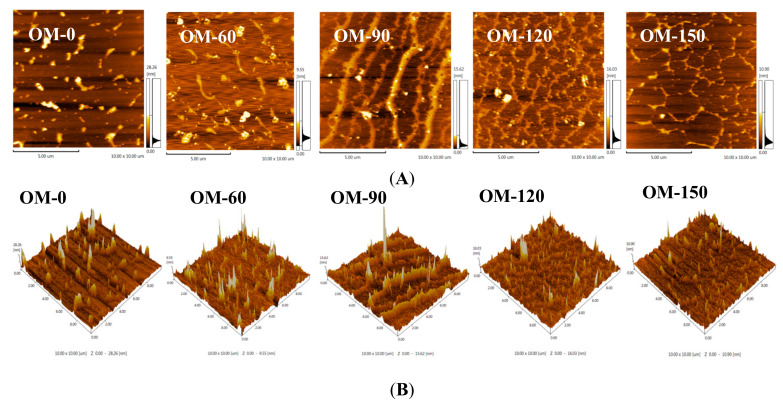
The 2D and 3D main morphologies of whole-grain oat pulp samples under different pressures treated using DHPM: (**A**) 2D main morphologies; (**B**) 3D main morphologies.

**Table 1 foods-12-02747-t001:** Effects of different DHPM treatment pressures on particle size of whole-grain oat pulp.

Particle Size/μm	D [4,3]	D [3,2]	D (0.1)	D (0.5)	D (0.9)	Span
OM-0	72.597 ± 0.857 ^a^	18.618 ± 0.138 ^a^	7.644 ± 0.046 ^a^	42.367 ± 0.665 ^a^	183.160 ± 1.933 ^a^	4.143 ± 0.033 ^e^
OM-60	28.176 ± 0.098 ^b^	8.378 ± 0.008 ^b^	3.794 ± 0.010 ^b^	11.496 ± 0.021 ^b^	82.587 ± 0.478 ^b^	4.897 ± 0.046 ^d^
OM-90	22.776 ± 0.283 ^c^	7.794 ± 0.026 ^c^	3.645 ± 0.014 ^c^	10.360 ± 0.037 ^c^	65.499 ± 1.069 ^c^	5.190 ± 0.091 ^c^
OM-120	19.542 ± 0.431 ^d^	7.42 ± 0.020 ^d^	3.532 ± 0.010 ^d^	9.734 ± 0.050 ^d^	54.052 ± 1.573 ^d^	5.970 ± 0.127 ^b^
OM-150	18.608 ± 0.510 ^e^	7.28 ± 0.021 ^e^	3.472 ± 0.011 ^e^	9.552 ± 0.063 ^d^	50.247 ± 1.670 ^e^	6.854 ± 0.123 ^a^

Values are presented as means ± standard deviations (*n* = 3). Different lowercase letters in each column indicate significant differences among processing techniques (*p* < 0.05).

## Data Availability

The data presented in this study are available on request from the corresponding author.

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
