# Peer review of "Effect of Dynamic High-Pressure Microfluidizer on Physicochemical and Microstructural Properties of Whole-Grain Oat Pulp"

_foods, 2023, doi:10.3390/foods12142747_

Round 1
Reviewer 1 Report
Please refer to the attached file for comments and recommendations.

It is imperative to prioritize a professional approach by ensuring that the manuscript undergoes thorough editing by a qualified English proofreader prior to its publication. During the editing process, it has come to our attention that there are notable errors in grammar and spelling within the manuscript. Additionally, some sentences have become excessively long and would benefit from concise and precise restructuring. Addressing these issues will contribute to maintaining a high standard of professionalism and readability in the final published work.
Author Response
Please see the attachment. The English-Editing Certificate is listed below.

Reviewer 2 Report
Manuscript is quite well prepared, brings new data on the possibility of preparation of whole oat grain drinks, without production of wastes. In my opinion work should be slightly corrected and supplemented. Briefly, I propose:
1) Information “Bovine serum albumin (BSA, > 98% pure, Sigma, USA)…” – please add to paragraph Materials,
2) Wet oat grinder – please add apparatus and parameters,
3) Dynamic high-pressure microfluidizer – please add producer, type, etc.,
4) Please delete “damaged starch assay kit” (page 2) or add these data,
5) NT-MDT Prima; Bruker Dimension Edge – please add instrument model,
6) Dispersion characteristics – please give apparatus,
7) For beta-glucan discussion – please add more discussion on reasons of it higher content in DHPM treated oat pulps (correlation with pressure); please give typical beta-glucan content (references),
8) Indacated (page 4) – please correct mistake,
9) Figure 1 – please add statistical caption,
10) Discussion for Figure 3 – please add information on oat starch granules sizes,
11) In Figures 5 and 6 – please correct axes captions.
In my opinion English language used in reviewed manuscript is quite good.
